# A Dynamic LLM-Powered Agent Network
# for Task-Oriented Agent Collaboration

**Zijun Liu**[1]*, **Yanzhe Zhang**[2], **Peng Li**[3], **Yang Liu**[1,3,4], **Diyi Yang**[5]
[1]Dept. of Comp. Sci. & Tech., Institute for AI, Tsinghua University, Beijing, China
[2]Georgia Institute of Technology, Georgia, USA
[3]Institute for AI Industry Research (AIR), Tsinghua University, Beijing, China
[4]Jiangsu Collaborative Innovation Center for Language Competence, Jiangsu, China
[5]Stanford University, California, USA
liuzijun20@mails.tsinghua.edu.cn, diyiy@stanford.edu

## Abstract

Recent studies show that collaborating multiple large language model (LLM) powered agents is a promising way for task solving. However, current approaches are constrained by using a fixed number of agents and static communication structures. In this work, we propose automatically selecting a team of agents from candidates to collaborate in a dynamic communication structure toward different tasks and domains. Specifically, we build a framework named Dynamic LLM-Powered Agent Network (**DyLAN**) for LLM-powered agent collaboration, operating a two-stage paradigm: (1) Team Optimization and (2) Task Solving. During the first stage, we utilize an *agent selection* algorithm, based on an unsupervised metric called *Agent Importance Score*, enabling the selection of best agents according to their contributions in a preliminary trial, oriented to the given task. Then, in the second stage, the selected agents collaborate dynamically according to the query. Empirically, we demonstrate that DyLAN outperforms strong baselines in code generation, decision-making, general reasoning, and arithmetic reasoning tasks with moderate computational cost. On specific subjects in MMLU, selecting a team of agents in the team optimization stage improves accuracy by up to 25.0% in DyLAN.[1]

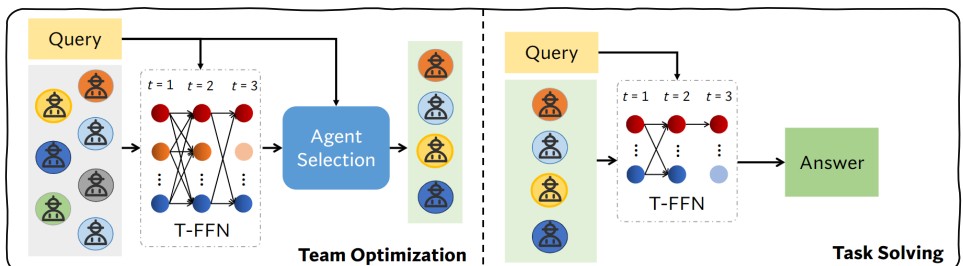

Figure 1: DyLAN adopts a two-stage paradigm. Agents communicate in a structure of the temporal feed-forward network (T-FFN). At the "Team Optimization" stage, DyLAN performs *agent selection* for the most contributory agents in a primary collaboration, oriented to tasks or domains. The selected agents then collaborate dynamically for an answer on the given query at the "Task Solving" stage.

---

*Work done when the first author was a UGVR visiting student at Stanford University.
[1]Code is available at https://github.com/SALT-NLP/DyLAN.

# 1  Introduction

Large Language Model (LLM) agents (Richards & et al., 2023; Nakajima, 2023; Reworkd, 2023) have demonstrated promising performance on various tasks, ranging from reasoning (Yao et al., 2023), code generation (Shinn et al., 2023) to embodied tasks such as video gaming (Wang et al., 2023a) and autopilot systems (Jin et al., 2023). Given the impracticality of a single agent managing all these tasks efficiently, recent research has shifted towards multi-agent collaborations, yielding significant advancements (Li et al., 2023; Du et al., 2023; Wang et al., 2023c; Jiang et al., 2023; Shinn et al., 2023; Chen et al., 2024; Wu et al., 2023).

As an analogy to human society, how human teams function may provide valuable insights for developing more effective multi-agent collaboration systems. For instance, recent studies have demonstrated that certain effective communication structures, derived from human society, also play a positive role in multi-agent collaborations (Yin et al., 2023; Chen et al., 2024). In addition to communication structures, another notable characteristic of human teams is that they would optimize team members according to the given task. Take medical consultations as an example. Collaborations in dynamic structures are evident when the composition of the team changes within the procedure of consultation, as some doctors may become less relevant in major as the conversation goes deeper and "leave" the consultation, leading to corresponding changes in the communication structure. Team optimization is often observed in the varying initial composition of doctors for consultations with different diseases, influenced by changes in the related medical fields and the contribution of each doctor. These characteristics prompt an important question: *Does a dynamically changing team of agents benefit LLM-powered agent collaborations similarly?*

However, the question is not well-addressed yet. While various communication structures have been studied for different tasks, such as debating for reasoning (Du et al., 2023; Liang et al., 2023; Xiong et al., 2023) and self-collaboration for coding (Dong et al., 2023; Qian et al., 2023a;b), these communication structures do not alter members in the agent team and remain fixed throughout the collaboration. It indicates that task-oriented dynamic selections in agents are not thoroughly explored in current research. Furthermore, in the context of agent teams, most existing studies opt for hand-crafting agents from human priors (Liu et al., 2023; Nakajima, 2023; Hong et al., 2024; Shinn et al., 2023; Li et al., 2023) or employ an LLM to generate them (Wang et al., 2023c; Chen et al., 2023b; Christianos et al., 2023). These approaches generally predefine agents without further validation of the collaboration process. This leads to static agent teams or rebuilding teams without principled verification (Chen et al., 2024). Challenges still remain for optimization methods.

As a first attempt towards addressing the above question, we introduce a novel framework named Dynamic LLM-Powered Agent Network (**DyLAN**). DyLAN conceptualizes multi-agent collaboration using temporal feed-forward networks (T-FFNs). In this formulation, each communication step of the agents corresponds to a network layer, with nodes representing the agents involved at that step and edges indicating communications between agents, for incorporating dynamic agent teams agnostically. DyLAN functions in two stages to incorporate task-oriented agent collaborations (Figure 1). The first stage is termed **Team Optimization**, where we select top contributory agents unsupervisedly among the initial team of candidates according to the task query, based on their individual contributions. We propose a forward-backward message passing algorithm on the T-FFN termed *agent selection* in Section 3.4, inspired by the back-propagation algorithm (Rumelhart et al., 1986) and neuron importance scores (Yu et al., 2018). This algorithm measures the contribution of each agent at the first stage with an unsupervised metric named *Agent Importance Score*. The most contributory agents form a smaller team to collaborate at the second stage — **Task Solving**, thereby minimizing the impact of less effective agents on the final answer. Specifically, the collaboration begins with a team of agents, and an LLM-powered ranker in the middle dynamically deactivates low-performing agents (i.e., *agent team reformation*), thus expanding the T-FFN, integrating dynamic communication structures into DyLAN (Section 3.3.2). Incorporating *agent selection*, DyLAN effectively identifies and coordinates a task-oriented team of agents in a principled way. Extensive experiments demonstrate that DyLAN outperforms strong baselines in various tasks, including code generation, decision-making, general reasoning, and arithmetic reasoning. Notably, *agent selection* in DyLAN has

improved accuracy by up to 25.0% in certain subjects of the MMLU dataset (Hendrycks et al., 2021a), underlining the significance of dynamic agent teams.

In summary, our contributions are threefold:

- We introduce a novel framework named DyLAN for task-oriented agent collaboration in two stages with *agent selection*, marking a significant advancement in the study of dynamic agent teams.
- DyLAN innovatively formulates agent collaborations in temporal feed-forward networks with *agent team reformation*, enhancing its adaptability and reducing dependence on human preconceptions.
- Empirical results demonstrate the superior accuracy, efficiency, and stability of DyLAN across various tasks, underscoring the need for dynamic agent teams.

## 2 Related Work

**Team Optimization of LLM-Powered Agents** The construction of agent teams is the essential and initial step for LLM-powered agent collaboration. TPTU (Ruan et al., 2023) and Chameleon (Lu et al., 2023) decompose tasks to choose or create tools accordingly. Recent studies also use LLMs to generate a fixed number of role prompts for agents in response to a task query (Wang et al., 2023c; Suzgun & Tauman Kalai, 2024), or for each round of discussion (Chen et al., 2024). However, manual prompts require careful design, which is impractical for adaptation on each task or domain, and prompting LLMs with predefined or generated descriptions may not result in the desired abilities of the agents without verification. Therefore, posteriorly selecting a team of agents based on their actual behaviors in the collaboration according to the task becomes essential. While team optimization for LLM agents is a relatively new area, human-team optimization has been studied for a long time. For instance, Liu et al. (2015) show that skill contribution is essential for selecting crowd workers to solve outsourced tasks efficiently. Based on peer rating, researchers have developed an algorithm for managing online workers in an optimal organization (Lykourentzou et al., 2022). Drawing inspirations, we introduce an unsupervised algorithm to select a team of agents by quantifying their contributions based on peer ratings in Section 3.4.

**Communication Structures in LLM-Powered Agent Collaboration** Collaboration between multiple LLM agents has demonstrated strong performance on various tasks in recent years and has emerged as a promising approach to enhance the capabilities of individual LLMs. To enable collaborations between multiple agents, recent studies have developed different communication structures and assigned agents in pre-defined architecture. For instance, researchers have found taking multiple LLM instances to debate for a fixed number of rounds can boost their factuality and reasoning capacities (Du et al., 2023; Liang et al., 2023; Xiong et al., 2023). To aggregate multiple LLM responses, LLM-Blender (Jiang et al., 2023) calls different LLMs in one round and uses pairwise ranking to combine the top responses. It has also been shown effective in distributing workloads to LLMs and concatenating their answers, thus producing better results (Ning et al., 2024; Suzgun & Tauman Kalai, 2024; Qiao et al., 2024). It is worth noting that existing studies (Hao et al., 2023; Zhang et al., 2023b) have tried organizing LLM instances into linear layers, but they mainly studied supervised learning in context space and LLM evaluation, respectively, not the scenario in which we are interested. However, running LLMs in a static architecture may limit the performance and generalization. On specific reasoning tasks, adopting a dynamic directed acyclic graph structure for LLMs has been shown effective (Zhang et al., 2023).Also, recent studies (Yin et al., 2023; Chen et al., 2024; Zhang et al., 2023a; Zhuge et al., 2024) have demonstrated that optimal communication structures vary with tasks and compositions of agents. Aligned with the findings, we propose a structure that adjusts dynamically based on selecting agents according to the tasks and the construction of the agent team in Section 3.3.2.

**Evaluation of the Contribution of LLM-Powered Agents** It is non-trivial to evaluate the contribution of each LLM agent in a multi-agent system, especially when they communicate over multiple rounds. In the single-round setting, existing methods use LLMs heavily for evaluation. To overcome the over confidence of LLMs (Xiong et al., 2024), pairwise

| Method | Single Exec. | LLM-Blender | LLM Debate | Reflexion | CAMEL | AgentVerse | DyLAN |
|---|---|---|---|---|---|---|---|
| **Communication Structure** $(\mathcal{V}; E)$ | | | | | | | |
| **Multiple Roles** | × | × | × | Manual | Manual | Generated | Man.&Gen. |
| **Early Stopping** | × | × | × | ✓ | ✓ | ✓ | ✓ |
| **Dynamic Structure** | ✓ | × | × | × | × | × | ✓ |
| **Team Optimization** | × | × | × | × | × | × | ✓ |

Table 1: Comparison between **DyLAN** and representative previous works. In the second row, nodes denote agents at different time steps ($\mathcal{V}$), arrows represent edges ($E$), and color indicates the role of agents.

ranking based on an additional LLM ranker has been introduced in LLM-Blender (Jiang et al., 2023). To rank $n$ responses with an independent LLM in a single round, they compare all $O(n^2)$ pairs. For better efficiency, researchers use a $k$-length sliding window to choose top $k$ responses within $O(nk)$ pairwise comparisons (Qin et al., 2023). However, these methods have not been extended to multi-round settings. Inspired by the neuron importance score (Yu et al., 2018), we evaluate agents by propagating and aggregating single-round peer ratings in a back-propagation manner (Rumelhart et al., 1986). In this way, we then introduce an unsupervised metric called *Agent Importance Score* to quantify the contribution of each agent in multi-round collaborations (Section 3.4).

# 3 Dynamic LLM-Powered Agent Network

## 3.1 Overview

We introduce a framework for LLM-powered agent collaboration named Dynamic LLM-Powered Agent Network (**DyLAN**), facilitating dynamic communication structures and automatically task-oriented *agent selection* in a two-stage fashion (Figure 1): an optimized agent team is constructed in the first stage "Team Optimization" through a preliminary trial and then the team collaborates to solve the task in the second stage "Task Solving".

A core component of DyLAN is the temporal feed-forward networks (T-FFNs), whose nodes denote agents and edges denote the communication channels between agents (Figure 2 left). T-FFNs serve not only as the abstraction of communication structures but also the computation graph. From this perspective, as shown in Table 1, various LLM-powered agent collaboration systems (Jiang et al., 2023; Shinn et al., 2023; Du et al., 2023; Li et al., 2023; Chen et al., 2024) can be represented by similar network structures as T-FFNs. DyLAN is the only framework that supports multiple agents with roles and tools, early stopping (Section 3.3.2), dynamic communication structures and team optimization simultaneously. To be specific, for team optimization, our *agent selection* algorithm is performed as a backward message passing algorithm on the T-FFN (Figure 2 right), and for task solving, *agent team reformation* expands the T-FFN dynamically with messages passing forward.

To make it easier to understand, we will start by explaining the formulation of T-FFNs. Then, we will move on to the task solving stage, and finally, we will explain the team optimization stage, which relies on components of task solving.

## 3.2 Temporal Feed-Forward Networks (T-FFNs)

A T-FFN is a multi-layer network, of which each layer represents a time step. Its formal definition is as follows.

**Definition 1 (Agents)** *Agents participating the collaboration are represented by*

$$\mathcal{A} = \{a_1, a_2, \cdots, a_N\}, \tag{1}$$

*where N denotes the total number of agents, and $a_i$ can be (I) an LLM-powered agent possibly equipped with tools, or (II) an independent tool, e.g., scripts, code interpreters.*

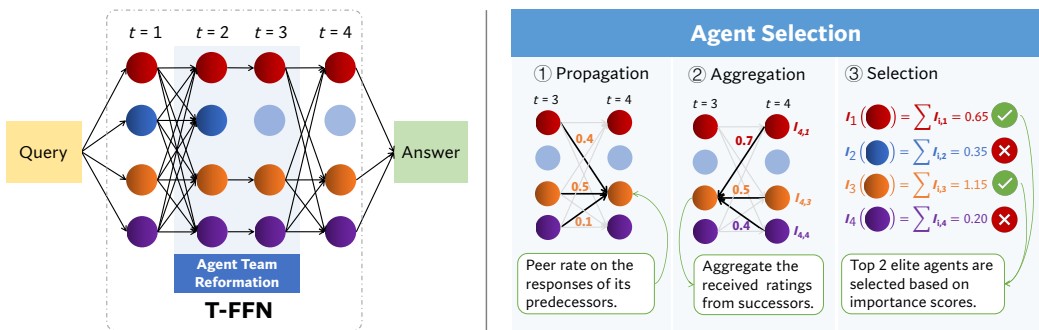

Figure 2: The left part shows how DyLAN outputs the answer in a temporal feed-forward network (T-FFN), where nodes represent agents at specific time steps (Section 3.2). Agent team reformation functions in the middle steps, during which the low-performing agent is deactivated in subsequent time steps. The right part depicts *agent selection* (Section 3.4), where the contribution of each agent in a primary trial is automatically evaluated in three steps using *Agent Importance Score*, denoted as $I$. Then, the top-ranked agents based on $I$ will be selected as the optimized, task-oriented team of agents.

**Definition 2 (Nodes)** *The t-th layer of a T-FFN consists of N nodes, each of which corresponding to one agent:*

$$\mathcal{V}_t = \{v_{t,1}, \cdots, v_{t,N}\}, \tag{2}$$

*where $t = 1, \cdots, T$, and node $v_{t,i}$ corresponds to agent $a_i$.*

**Definition 3 (Edges)** *Edges in a T-FFN refer to the communication channels between nodes, forming the communication structure between agents. Specifically, the set of edges between the nodes in layer $t-1$ and $t$ is denoted as*

$$E_{t-1,t} = \{(v_{t-1,i}, v_{t,j})\} \subseteq \mathcal{V}_{t-1} \times \mathcal{V}_t, \tag{3}$$

*where $t = 2, \cdots, T$, and $(v_{t-1,i}, v_{t,j})$ denotes an edge connecting nodes $v_{t-1,i}$ and $v_{t,j}$.*

**Definition 4 (T-FFN)** *Finally, the T-FFN corresponding to the collaboration is defined as a T-layer network:*

$$\mathcal{G} = (\mathcal{V}_1, \cdots, \mathcal{V}_T; E_{1,2}, \cdots, E_{T-1,T}). \tag{4}$$

*Note that we only consider T-FFNs where edges only exist in adjacent layers. However, edge can be added to arbitrary pairs of nodes, making the T-FFN capable of representing more complex communication structures.*

## 3.3 Task Solving

Task solving involves performing *inference* on the T-FFNs, jointly with *agent team reformation*, which be elaborated on in the subsequent two sections.

### 3.3.1 Inference

Before details, we first introduce the formulation of message passing on T-FFNs.

**Definition 5 (Message Passing)** *Given a T-FFN, a node $v_{t,j}$, a set of adjacent nodes $\mathcal{U} = \{u_1, \cdots, u_K\}$, and the messages $\mathcal{M} = \{m_{u_1}, \cdots, m_{u_K}\}$ received by $v_{t,j}$, where K is the size of $\mathcal{U}$, and $m_{u_k}$ is the message sent from $u_k$ to $v_{t,j}$, message passing aggregates all the messages $\mathcal{M}$ to produce a updated message $\hat{m}_{v_{t,j}}$ for $v_{t,j}$, which is formally defined as*

$$\hat{m}_{v_{t,j}} = f_{\mathrm{mp}}\left(\mathcal{M}, v_{t,j}\right), \tag{5}$$

*where $f_{\mathrm{mp}}(\cdot, \cdot)$ is the aggregator function.*

**Definition 6** *We refer the algorithm as* **forward message passing** *when $\mathcal{U}$ is the set of all adjacent nodes of $v_{t,j}$ from the previous time step, i.e., $\mathcal{U} = \{u_k | \forall u_k, (u_k, v_{t,j}) \in E_{t-1,t}\}$. Similarly, it is referred as* **backward message passing** *when $\mathcal{U}$ is the set of all adjacent nodes from the next time step: $\mathcal{U} = \{u_k | \forall u_k, (v_{t,j}, u_k) \in E_{t,t+1}\}$.*

With the above formulation, we can describe the inference process of a T-FFN $\mathcal{G}$ in the manner of **forward message passing**. During collaborations on a given task, an agent at a specific time step takes the responses, i.e., messages, from other agents at the previous time step as input and generates responses based on the task query. Based on different types of agents at $v_{t,j}$, we can implement $f_{\mathrm{mp}}(\cdot, v_{t,j})$ respectively: (I) concatenating input messages along with the task query into prompt templates (refer to task instructions in Appendix D) and take the response from LLM after generation or tool calling, or (II) filter the input that the tool can process, e.g., code completions and structured text.

During inference, we begin feeding the task query $q \in \mathcal{Q}$ into agents at time step 1 ($\mathcal{V}_1$), where $\mathcal{Q}$ denotes the dataset. By passing responses of nodes $\mathcal{V}_{t-1}$ at time step $t-1$ to nodes $\mathcal{V}_t$ at $t$, agents can perceive responses from all nodes at the previous time step and perform collaborative behavior, which might include criticizes, advice, refinement, or quality reviews, depending on the implementation of agents. Formally, the inference process is defined as

$$f_{\mathrm{Infer}}(q, \mathcal{G}) = o, \tag{6}$$

where $o = \mathrm{argmax}\{\mathcal{M}_T\}$ and $\mathcal{M}_T$ denotes the responses from $\mathcal{V}_T$. Please refer to Algorithm 1 for detailed procedure.

### 3.3.2 Agent Team Reformation

Given a set of agents $\mathcal{A}$, *agent team reformation* aims to identify more contributory agents and construct a dynamic communication structure accordingly. To this end, we leverage an additional LLM instance, referred as the "LLM Ranker", to analyze responses from the agents participate in the current time step and give out a ranking, prompted by the template of "Ranker" in Appendix D. Then, the top-ranked agents are allowed to participate in the next time step. In other words, edges will only be added for these top-ranked agents, resulting in a dynamic communication structure.

Formally, suppose the set of agents participates in time step $t$ is $\mathcal{A}_t = \{a_k\}$, and the top-ranked agents are $\mathcal{A}_{t+1} = \{a_l\}$, where $k$ and $l$ are the indices of the agents as defined in Equation (1), then we can obtain two nodes sets $\mathcal{V}'_t = \{v_{t,k} | \forall a_k \in \mathcal{A}_t\}$ and $\mathcal{V}'_{t+1} = \{v_{t+1,l} | \forall a_l \in \mathcal{A}_{t+1}\}$, and the edge set $E_{t,t+1}$ is defined as

$$E_{t,t+1} = \mathcal{V}'_t \times \mathcal{V}'_{t+1}. \tag{7}$$

The process progresses iteratively until the stop condition is met, and finally, we get a T-FFN $\mathcal{G}^{\mathcal{A}}_q$ for the query input $q$. We use function $f_{\mathrm{IAS}}$ to denote the entire computation:

$$\mathcal{G}^{\mathcal{A}}_q = f_{\mathrm{IAS}}(\mathcal{A}, q). \tag{8}$$

To further enhance efficiency, we introduce an **early-stopping mechanism**. Inspired by the Byzantine Consensus theory (Castro & Liskov, 1999), at least $3p + 1$ agents are needed to tolerate $p$ faulty agents in a single round of communication. Following the theory, the inference process will be terminated when over $2/3$ of agents in a single layer have a consistent answer. In practice, the inference process will also be terminated when the maximum time step is reached. Note that none of the consistency measures used in prior work (Wang et al., 2023b; Aggarwal et al., 2023; Yin et al., 2023) applies to multi-round multi-agent interaction since their theories are assumed to execute a single LLM instance multiple times or expect all agents to reach the same answer.

### 3.4 Team Optimization

The goal of team optimization is to select a subset of agents from candidates based on their contributions evaluated from a primary trial, such that the new team solves the task query

| Method | Pass@1 | | #API Calls |
|---|---|---|---|
| Single Execution | 73.2 | (+0.0) | 1.00 |
| CodeT | 65.8 | (-7.4) | 20.00 |
| CodeT (Codex) | 74.8 | (+1.6) | 20.00 |
| Reflexion | 68.3 | (-4.9) | 4.05 |
| LATS | 81.1 | (+7.9) | 48.00 |
| CAMEL | 69.5 | (-4.1) | 12.03 |
| AgentVerse | 75.0 | (+1.8) | 22.50 |
| **DyLAN** (*Ours*) | **82.9** | (+9.7) | 16.85 |

| Method | Reward | | Success Rate | #API Calls |
|---|---|---|---|---|
| Direct Execution | 50.6 | (+0.0) | 28.0 | 14.52 |
| ReAct | 53.8 | (+3.2) | 30.0 | 8.40 |
| ReAct-SC | 58.0 | (+7.4) | 36.0 | 25.75 |
| Reflexion (*trial=4*) | 62.0 | (+11.4) | 40.0 | 25.40 |
| LATS | 64.5 | (+13.9) | 38.0 | > 400 |
| BOLAA | 66.0 | (+15.4) | 40.0 | 32.40 |
| **DyLAN** (*Ours*) | **68.3** | (+17.7) | **42.0** | 24.85 |

Table 2: Experimental results on the CG task (left) and results on the DM task (right). The number in parentheses indicates the difference relative to the single execution or direct execution. We indicate the foundation model of methods except for GPT-35-turbo. The median of three trials is reported when non-zero `temperature` is used.

more effectively and efficiently. Formally, given a task query $q$, a set of agents $\mathcal{A}$, a trial is performed based on the algorithm proposed in Section 3.3.2 resulting in a T-FNN $\mathcal{G}_q^{\mathcal{A}}$. And team optimization is formulated as

$$\hat{\mathcal{A}} = f_{\text{Optim}}(\mathcal{A}, \mathcal{G}_q^{\mathcal{A}}, q), \text{ where } \hat{\mathcal{A}} \subset \mathcal{A}. \tag{9}$$

$f_{\text{Optim}}$ is implemented as in a three-step procedure of *agent selection* (Figure 2 right):

(1) *Propagation*: Each node rates the solutions to the task query from its predecessors, which is a forward message passing process. Formally, given a node $v_{t,j}$ and an edge $(v_{t-1,i}, v_{t,j})$, the message $m_{v_{t-1,i}}$ sent from $v_{t-1,i}$ to $v_{t,j}$ is defined as the response to the task query $q$ from the agent $a_i$ at the previous time step. The aggregator function $f_{\text{mp}}(\cdot, v_{t,j})$ is implemented as a scoring function $f_{t,j}^{(s)}(\cdot, \cdot, \cdot)$, which maps the prompt $p_j$, the input query $q$, and all the messages $\mathcal{M}$ to the rating scores. Here, we use $w_{t-1,i,j}$ to refer to the rating score on $v_{t-1,i}$ from $v_{t,j}$, and

$$[w_{t-1,1,j}, w_{t-1,2,j}, ..., w_{t-1,N,j}] = f_{t,j}^{(s)}\left(p_j, q, \mathcal{M}\right). \tag{10}$$

(2) *Aggregation*: Each node aggregates the ratings it has received from its successors towards itself to quantify its own contribution independently at different time steps. The contribution of node $v_{t-1,i}$ is the sum of its successors' contribution multiplied by their peers' ratings on the agent's response. Aggregation is a backward message passing process. Formally, given a node $v_{t-1,i}$ and an edge $(v_{t-1,i}, v_{t,j})$, the message $m_{v_{t,j}}$ sent from $v_{t,j}$ to $v_{t-1,i}$ is defined as $w_{t-1,i,j}$. And the aggregator function $f_{\text{mp}}$ is defined as a weighted sum function:

$$\mathbf{I}_{t-1,i} = \sum_{(v_{t-1,i}, v_{t,j}) \in E_{t-1,t}} \mathbf{I}_{t,j} \cdot w_{t-1,i,j}, \tag{11}$$

where $\mathbf{I}_{t,i}$ denotes the contribution of $a_{t,i}$.

(3) *Selection*: During the last step, we sum up the scores for the same agent over all time steps to derive an importance score for each agent, and extract the top-$k$ agents that are most contributory according to these scores to form the optimized agent team. Formally, the *Agent Importance Score* $\mathbf{I}_i$ for agent $a_i$ is defined as

$$\mathbf{I}_i = \sum_{t=1}^{T} \mathbf{I}_{t,i}. \tag{12}$$

In practice, we initialize the contributions in the final layer first, and step backward to perform *Aggregation* layer by layer (Algorithm 2). The definition guarantees that the agent importance scores add up to 1 in each layer, which benefits fair comparison. Other details, such as initializing contributions in the final layer, are presented in Appendix B.2.

| Method | Prompting | Algebra | Counting and Probability | Geometry | Intermediate Algebra | Number Theory | Pre-Algebra | Pre-Calculus | Overall | #API Calls |
|---|---|---|---|---|---|---|---|---|---|---|
| Single Execution | CoT | 43.6 | 29.3 | 21.5 | 15.8 | 30.0 | 48.9 | 16.5 | 31.6 (+0.0) | 1.00 |
| LLM-Blender | | 47.5 | 25.5 | 23.8 | 13.8 | 39.7 | 46.7 | 15.8 | 31.7 (+0.1) | 6.00 |
| LLM Debate | | 50.2 | 25.3 | 22.3 | 13.1 | 28.9 | 48.0 | 19.0 | 32.4 (+0.8) | 8.00 |
| **DyLAN** (*Ours*) | | 52.9 | 27.2 | 25.3 | 15.5 | 33.5 | 55.2 | 19.0 | **35.7** (**+4.1**) | 7.15 |
| Single Execution | Complex CoT | 49.1 | 29.7 | 22.3 | 14.6 | 33.4 | 53.8 | 16.8 | 34.1 (+0.0) | 1.00 |
| PHP | | 51.1 | 33.7 | 25.4 | 17.1 | 35.1 | 57.7 | 16.1 | 36.5 (+2.4) | 3.67 |
| **DyLAN** (*Ours*) | | 53.7 | 33.3 | 26.1 | 18.1 | 33.5 | 58.7 | 18.9 | **37.6** (**+3.5**) | 6.21 |

Table 3: Accuracy (%) on the AR task. The number in parentheses indicates the performance difference relative to a single execution.

| Method | Hum-anities | Social Science | STEM | Other | Overall | #API Calls |
|---|---|---|---|---|---|---|
| Random | 25.0 | 25.0 | 25.0 | 25.0 | 25.0 | - |
| Single Exec. | 59.8 | 74.0 | 62.9 | 71.8 | 66.4 (+0.0) | 1.00 |
| LLM-Blender | 60.4 | 75.2 | 66.3 | 70.7 | 67.3 (+0.9) | 6.00 |
| LLM Debate | 59.8 | 77.4 | 69.0 | 75.5 | 69.3 (+2.9) | 12.00 |
| **DyLAN** | 62.1 | 79.1 | 69.7 | 75.5 | **70.5** (**+4.1**) | 4.39 |

Table 4: Accuracy (%) on the GR task. "Other" stands for subjects like business, health, and misc in the MMLU dataset. We report the median of three runs for experiments.

| Task | #Agents | Tool Usage | Performance Improvement | #API Calls |
|---|---|---|---|---|
| CG | 12 → 8 | ✓ | 76.2 → 82.9 | 23.04 → 16.85 |
| DM | 8 → 4 | ✗ | 53.0 → 68.3 | 32.03 → 24.85 |
| GR | 7 → 4 | ✗ | 69.5 → 70.5 | 8.30 → 4.39 |
| AR | 4 | ✗ | - | - |

Table 5: Demonstration of experiment settings, including the number of agents and the performance throughout team optimization. We report reward for the DM task.

# 4 Experiments

## 4.1 Setup

**Code Generation (CG)** We use the HumanEval benchmark, with 164 human-labeled function-level completion codes and unit tests (Chen et al., 2021). Unit tests are used to validate the correctness of generated codes. We leverage two strong baselines CodeT (Chen et al., 2023a) and Reflexion (Shinn et al., 2023) along with the single execution. For multi-agent baselines, we re-implement CAMEL (Li et al., 2023) and AgentVerse (Chen et al., 2024) under their original configurations for fair comparisons.

**Decision Making (DM)** We evaluate our methods in the WebShop environment, selecting 50 environments in its test set (Chen et al., 2021) following the setting of LATS (Zhou et al., 2023). WebShop requires to find the item given an instruction of the customer. It provides "reward" as an intrinsic metric for item-instruction relevance, and "success" is marked when the reward is 1.0. Besides ReAct (Yao et al., 2023) and Reflexion, we re-ran a multi-agent method BOLAA (Liu et al., 2023), and a single-agent method LATS as strong baselines.

**General Reasoning (GR)** For the general reasoning task, we use the MMLU dataset (Hendrycks et al., 2021a), which contains four categories of a vast amount of problems in 57 subjects. We down-sample 1/5 of the problems in the test set because of its huge quantity. We choose LLM Debate (Du et al., 2023), LLM-Blender (Jiang et al., 2023), and the single execution on LLM as baselines.

**Arithmetic Reasoning (AR)** We leverage the test set of MATH (Hendrycks et al., 2021b) for evaluation, which consists of 7 subareas and 5,000 questions in total. To draw a fair comparison and verify the robustness, we categorize methods by different prompting strategies and select strong baselines accordingly. Preliminary experiments show that collaborating agents in different domains (e.g., algebra and geometry experts) does not make significant improvement, therefore we adopt agents with same prompts for all methods.

**DyLAN Setup** In Table 5, we elaborate the setup of DyLAN. To keep in line with baseline methods, we only equipped DyLAN with code interpreters as tools in the CG task. It is worth noting that *agent selection* is performed for each subject in the GR task, for each web page in the DM task, and directly for CG task in the team optimization stage. Due to space limitations, please refer to Appendix B.1 for details.

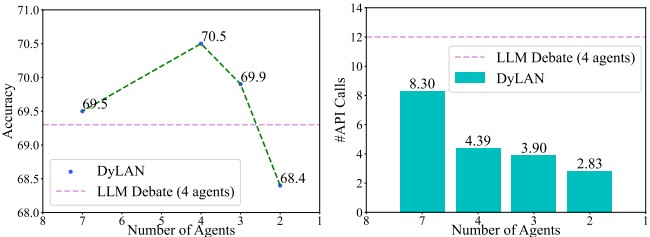

Figure 3: Impact of optimized agent team size. 2∼4 agents are selected from 7 candidate agents based on *Agent Importance Score*. Accuracy (left) and #API calls (right) on the GR task are visualized.

Table 6: Impact off the early-stopping mechanism (*es*) and agent team reformation (*atr*).

| Method | AR | | GR | |
|---|---|---|---|---|
| | Acc. | #API | Acc. | #API |
| **DyLAN** | **35.7** | **7.15** | **70.5** | **4.39** |
| *w/o es* | 35.0 | 13.00 | 70.1 | 13.00 |
| *w/o atr* | 33.8 | 8.20 | 69.9 | 7.05 |

| Method | CG | | DM | |
|---|---|---|---|---|
| | Pass@1 | #API | Reward | #API |
| **DyLAN** | **82.9** | **16.85** | **68.3** | **24.85** |
| *w/o es* | 80.5 | 19.00 | 67.5 | 54.25 |
| *w/o atr* | 76.2 | 17.98 | 66.0 | 48.90 |

## 4.2 Main Results

In Table 2, Table 3, and Table 4, we report the results on each dataset respectively. The number of API calls serves as a proxy for the efficiency of communication structures for agents, which cannot be clearly determined from token consumption which varies greatly depending on tasks and prompting strategies. Since "Task solving" after "Team Optimization" is essential for testing and deployment, we mainly report the cost from the second stage. The difference can be seen in Table 5, and further discussions in Appendix C.1.

**DyLAN improves overall performance on different tasks with a reasonable computational cost.** From Table 3, we find DyLAN realizes an 10.2% improvement to LLM Debate in terms of accuracy, with 10.6% lower #API calls (L3 vs. L4), suggesting it is a better trade-off between efficiency and effectiveness. Similar trends can be observed as DyLAN has a better performance with only 36.6% API calls of LLM Debate (L5 vs. L4 in Table 4), and 35.1% of LATS on CG and <6% on DM (L8 vs. L5 (left), L7 vs. L5 (right) in Table 2). We argue it can be attributed to the feed-forward structure and early-stopping mechanism, which allows different solutions to be delivered simultaneously and confirmed rapidly. In contrast, for methods in sequential architecture like PHP and Reflexion (Table 2), incorrect intermediates might easily influence the final output due to the single thread of reasoning or code generation and review. Also, ReAct on DM tasks exhibits similar failures due to misoperations in the middle. In contrast, Reflexion and LATS access the environment at certain states for multiple times to for reflection, limiting generalizabilities. In our case, any feedback from predecessors could be rated by successor nodes, making it easier to rectify potential invalid actions. Moreover, we see that DyLAN dynamically adjust the cost based on the difficulty of tasks. For instance, most questions in the MMLU dataset are less challenging than MATH, DyLAN has 2.76 fewer API calls on the query from the former. However, compared to other tasks, DyLAN introduces relatively lower improvements in AR tasks, which might be due to the high knowledge dependency of the MATH dateset.

**DyLAN benefits from the team optimization.** Moreover, we found that a dynamic team of task-oriented agents could enhance DyLAN. For different subjects in GR tasks, agent compositions are adjusted correspondingly to improve up to 25.0% in accuracy, as shown in Table 7. As denoted in Table 5, a dynamically selected team of agents could result in significant performance improvement, especially for DM tasks, where agents might have great interference from others. The overall performance can be significantly improved (up to 6.7%) with lower computational costs on each tasks after *agent selection*. Moreover, it also suggests that *Agent Importance Scores* can effectively capture and reflect the actual contributions of agents on a wide range of tasks. We further verify this claim in Appendix C.6.

## 4.3 Ablation Studies

**Impact of Optimized Agent Team Size** Fewer proper agents in a team could have better performance. As shown in Figure 3, DyLAN with an optimized team of 3 agents can outperform both DyLAN before team optimization and LLM Debate with 4 agents, suggesting the effectiveness of our proposed agent selection. The efficiency is also significantly improved

| Subject | Optimized Composition | Performance Improvement |
|---|---|---|
| college mathematics | Economist, Lawyer, Programmer, Mathematician | **25.0** : 40.0 → 65.0 |
| management | Lawyer, Psychologist, Economist, Programmer | **14.3** : 76.2 → 90.5 |
| high school statistics | Historian, Programmer, Psychologist, Mathematician | **9.3** : 65.1 → 74.4 |
| clinical knowledge | Doctor, Mathematician, Programmer, Psychologist | **5.7** : 69.8 → 75.5 |
| public relations | Historian, Psychologist, Lawyer, Mathematician | **4.5** : 54.5 → 59.1 |

Table 7: The optimized composition of agents and performance improvement on different subjects in the GR task.

| #Code Writers | #Code Reviewers | Pass@1 | #API Calls |
|---|---|---|---|
| 6 | 6 | 76.2 | 23.04 |
| 4 | 4 | 82.9 | 16.85 |
| 4 | 3 | 81.1 | 14.64 |
| 4 | 2 | 77.4 | 12.50 |
| 3 | 3 | 78.0 | 11.73 |
| 3 | 2 | 75.6 | 9.60 |

Table 8: Different compositions of agents on the CG task of an optimized team of agents. Agent teams are optimized by the *Agent Importance Score* from the first line.

by 52.9% and 67.8%, respectively. Probably because the imbalance of agents' expertise and opinions interfere with each other before optimization, especially on GR, where few candidates are relevant to a subject.

**Robustness of Agent Importance Score** The *Agent Importance Score* is robust over the imbalance of agent roles. On GR tasks, the candidates are imbalanced in terms of expertise. For most queries, there are less than 2 candidates that are related according to their role prompts. In Table 7, we found *agent selection* is capable for selecting related agents, e.g., "Mathematician" for "college mathematics", that matches human priors. However, if candidates are all vastly different from the task domain, e.g., "public relation", where the improvement is less significant. We also tested DyLAN on CG tasks with different amount of code writers and code reviewers after the "Team Optimization" stage. It is worth noting that a single run of "Team Optimization" could provides reusable *Agent Importance Scores* for multiple trials of agent selection. In Table 8, we exhibit the results under imbalanced teams of agents. We verified the imbalance of code writers and reviewers after optimization won't cause great performance drops. Though, reviewers affect the performance slightly greater than writers (L2,3 vs. L4,5), indicating the necessity of the amount of reviewers for code refinement.

**Impact of Early-Stopping and Agent Team Reformation** As shown in Table 6, early-stopping mechanism boosts efficiency to a great extent by minimizing #API calls by 45.0%, 66.2%, 11.3%, and 54.2% on AR, GR, CG, and DM tasks respectively, while providing slight performance improvement. Agent team reformation, however, is critical to enhance the correctness of the final answer. We conjecture it is because agents are filtered for temporary mistakes in LLMs, such as hallucinations, etc. Additionally, answer comparison is more challenging for open-ended tasks like CG or DM tasks. We use the BLEU score with a 0.9 threshold for consistency checks. This makes early stopping less effective since agents may generate codes in different formats, leading to fewer opportunities to stop early.

**Stability of DyLAN with Different Backbone Models** There is also a notable difference in CG tasks when the backbone model changes (Table 2). Reflexion and CodeT's performances are heavily related to the backbone model (L4 vs. L5 and L6 vs. L9). Instead, DyLAN shows a steady, consistent high performance (L7 vs. L10) under different backbone models with almost the same amount of API calls.

## 5 Conclusion and Future Work

This work introduces a framework named Dynamic LLM-Powered Agent Network (DyLAN) for collaboration of dynamic agent teams on complicated tasks. DyLAN functions in a two-stage paradigm, enabling agents to interact in a dynamic structure with agent team reformation. In "Team Optimization" stage, the *agent selection* algorithm based on an unsupervised metric termed *Agent Importance Score*, selects top contributory agents in a principled way for collaboration on "Task Solving". Overall, DyLAN reveals improvement on diverse tasks with relatively less computational cost compared to baselines. In the future, we plan to explore the effectiveness of DyLAN built on open-source foundation models.

**Acknowledgments**

We thank Yilun Du for the helpful assistance on code implementation. We sincerely thank William Held, Ruibo Liu, Dr. Yue Zhuge, Noah Shinn and Kangfu Zheng for their valuable feedback on the project.

## Ethics Statement

LLM-powered agent systems are widely used in practical applications. DyLAN could also effortlessly cover practical software development, virtual room chat, video games, and so on (Hong et al., 2024; Nascimento et al., 2023; Zhou et al., 2023; Zhu et al., 2023; Chan et al., 2024). In these open-world environments, agents may operate as planners, actors, etc. DyLAN only requires people to give rough instructions on the constitute of agents and could automatically optimize a better team of agents to construct an efficient multi-agent system. These systems could benefit from DyLAN to reduce human labor on designing agents and have a better performance on their target tasks.

Also, the overall architecture of DyLAN (Figure 2) reflects the optimal collaboration organization of human online workers (Lykourentzou et al., 2022), and reveals significant performance in agent collaborations. Therefore, simulating human collaboration by LLM-powered agent collaborations under DyLAN might also be possible. Optimizing human collaboration by searching and simulating LLM agents will hopefully be more convenient and effective. We also acknowledge the potential risk in pretrained language models used in the paper, e.g., GPT-3.5 and GPT-4, which may cause improper responses. Furthermore, creating agents by hand and LLM generations might prompt LLM-powered agents to act or response misaligned with principles of society, which may possibly happen during agent collaborations. However, we think team optimization process could potentially alleviate this situation.

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

## A    Discussion & Limitation

In experiments, we view code generation tasks as representative of open-ended generation tasks and adopt BLEU to decide whether two answers are consistent in early stopping mechanism in Section 3.3.2. In fact, the performance could be further leveraged by task-specific methods like CodeBLEU (Ren et al., 2020) or CodeT (Chen et al., 2023a).

For practical usage, the agent-evaluation metrics could cooperate with human annotation to give a more precise evaluation result on individual contributions of agents, mainly when facing data scarcity problems. Furthermore, we simply incorporating *agent selection* on Dy-LAN with *agent team reformation*, as a primary step towards collaboration of dynamic agent teams. It still remains to be seen how to cooperate off-collaboration and in-collaboration optimization methods in a finer granularity to further improve performance and efficiency in LLM-powered agent collaboration systems.

Additionally, though *agent selection* could differentiate top contributory agents, in extreme cases where the majority of agents are designed to contradict the task requirement, low performance might be caused, e.g., agents are prompted or trained to generate codes may face difficulties in clinical question answering. To tackle the imbalance of high- and low-performing agents, replicating agents with high *Agent Importance Score* instead of including low-score agents could be a solution. Additionally, in extreme circumstances, we can automatically introduce agents from more capable LLMs with validation, in addition to *agent selection*.

## B    Implementation Details

### B.1    Detailed Experiment Settings

---

**Algorithm 1** The Inference Process of **DyLAN** on an Arbitrary Query

---

**Input:** T-FFN $\mathcal{G} = (\mathcal{V}_1, \cdots, \mathcal{V}_T; E_1, \cdots, E_{T-1,T})$, Query $q$
**Output:** Final Answer $o$
// $E = \{(v_{t,i}, v_{t+1,j})\}_{t=1}^{T-1}, v_{t,i}, v_{t+1,j} \in \mathcal{V}$
//    $= \bigcup_{t=1}^{T} \mathcal{V}_t$.
// $m_{t,i} \in \mathcal{M}_t$ denotes the response from $v_{t,i} \in \mathcal{V}_t$.

**for** $t = 1; T$ **do**
  **if** *agent team reformation* at time step $t$ **then**
    $\mathcal{M}_{top} \leftarrow \text{top}-k(\{m_{t-1,j}|v_{t-1,j} \in \mathcal{V}_{t-1}\})$
    $E \leftarrow E \backslash \{(v_{t',j}, *), (*, v_{t',j})|m_{t',j} \in \mathcal{M}_{top}, t' \geq t-1\}$
    $m_{t,j} \leftarrow m_{t-1,j}, \forall (v_{t-1,j}, v_{t,j}) \in E$
  **else**
    $\forall i, \exists k, (v_{t,i}, v_{t+1,k}) \in E_{t,t+1}$,
    $m_{t,i} \leftarrow f_{mp}(\{m_{t-1,j}|(v_{t-1,j}, v_{t,i}) \in E_{t,t+1}\}, v_{t,i})$
  **end if**
  **if** early stopping **then**
    $T \leftarrow t$
    break
  **end if**
**end for**
$o \leftarrow \text{postProcess}(\text{maxCount}\{\mathcal{M}_T\})$

---

**Algorithm 2** The Team Optimization Process $f_{\text{Optim}}$ of *Agent Importance Score* within **DyLAN**

---

**Input:** Output $o$, T-FFN $\mathcal{G} = (\mathcal{V}, E)$
**Output:** *Agent Importance Score* of agents $I$
// $m_{t,i} \in \mathcal{M}_t$ denotes the response from $v_{t,i} \in \mathcal{V}_t$.
flag $\leftarrow$ False
**for** $t = T; 1$ **do**
  **if** $\{v_{t,i}|\exists k, (v_{t-1,k}, v_{t,i}) \in E\} \neq \phi$ **then**
    **if** $\neg$flag **then**
      flag $\leftarrow$ True
      distribute scores for $I_{t,i}$
    **else**
      $\mathcal{M}_{t-1} \leftarrow \{m_{t-1,j}|(v_{t-1,j}, v_{t,i}) \in E\}$
      $[w_{t-1,1,i}, ..., w_{t-1,m,i}] \leftarrow f_{t,i}^{(s)}(p_i, q, \mathcal{M}_{t-1})$
      $I_{t-1,j} \leftarrow I_{t-1,j} + I_{t,i} w_{t-1,j,i}$, $\exists (v_{t-1,j}, v_{t,i}) \in E$
    **end if**
  **end if**
**end for**

---

**Common Settings** In all experiments, we use `gpt-35-tubo-0301` for every method if not specified. The version of GPT-4 is `GPT-4-0613`. In Table 2, "(Codex)" denotes `code-davinci-002` from OpenAI (Chen et al., 2021; OpenAI, 2023). All experiments with non-zero `temperature` is repeated for three times and the median is reported. To avoid the context length issue in prior work (Du et al., 2023; Liu et al., 2024), we set memory space for agents in DyLAN to 1 only to keep the freshest responses of predecessors. We set max tokens to 2048 for GR and AR tasks and 1024 for CG and DM tasks to avoid exceeding the maximum context length. The construction of candidates are demonstrated in Appendix D. We set $N = 4$ in T-FFN after team optimization because the early-stopping mechanism requires at least four agents to tolerate one different response at a specific time step (Section 3.3.2); when reaching consensus over 2/3 of agents, it allows for $4 - \lceil \frac{2}{3}N \rceil = 1$ excetional response. We use a listwise ranker in the agent team reformation of DyLAN because of the effectiveness and efficiency, compared to ELo rating (Herbrich et al., 2006) or Sliding Window (Qin et al., 2023) we have tested in Appendix C.5. We use the same ranker to implement LLM-Blender (Jiang et al., 2023) in experiments. We set $k = 2$ in the *agent team reformation*, because it's the minimal number for collaborations and we empirically found it brings great trade-off between effectiveness and efficiency. To avoid positional bias, for each time step $t$, we shuffle the responses from agents at $t - 1$ when passing messages towards agents at $t$. The detailed inference algorithm is in Algorithm 1. To implement the early-stopping mechanism, we need to determine whether the answers from the nodes in the same layer of DyLAN are consistent. For classification and decision-making problems, the answers are consistent if identical, and for open-ended generation, the consistency is determined by a threshold of `BLEU` score.

**Experiments on Reasoning Tasks** In general reasoning, we extract the answer from the response by matching the last "(X" or "(X)", where "X" represents A, B, C or D. On average, Agent team reformation functions on the third time step. They could go through at maximum $T = 4$ rounds of interaction. We also searched `temperature` in $\{0, 0.2, 0.8, 1.0\}$ for the best configuration for each system. In arithmetic reasoning, we set `temperature` to 0 for the single execution and PHP, 0.2 for LLM Debate, LLM-Blender, and DyLAN with Complex CoT prompts, and 1.0 for DyLAN with simple CoT prompts in Table 3, since systems with the same prompts will give all the same responses if `temperature` is zero, causing degradation. Prompting templates are replicated from their original studies, including normal CoT prompts (Wei et al., 2022) from the MATH dataset (Hendrycks et al., 2021b) and Complex CoT from PHP (Zheng et al., 2023). We follow the answer extraction method from the origin paper (Hendrycks et al., 2021b). We construct DyLAN with 4 agents assigned no specific roles and let agents to interact for at maximum $T = 4$ rounds under T-FFN formulation. We reported the classification accuracy of each category averaged across subjects and the numbers of API calls of running DyLAN on the optimized team of agents.

**Experiments on Code Generation Tasks** In the code generation task, we set `temperature` to 0 for the single execution, Reflexion, and 0.8 for LLM Debate, LLM-Blender, CodeT, and DyLAN in Table 2. In DyLAN, we optimized four agents to write code and four agents to give code reviews from 12 candidates in Appendix D. The selected code writers are "Python Assistant", "Algorithm Developer", "Computer Scientist", and "Programmer"; and the selected code reviewers are "Syntax Checker", "Unit Tester", "Reflector", and "Ranker". "Syntax Checker" is pure external tools using a code interpreter for syntax checking without LLMs, and "Unit Tester" is equipped with a code interpreter. The tool is triggered when LLM generated codes inside the format ```python\n(code)\n```. In DyLAN, solutions given by code writers are reviewed by code reviewers in at maximum $T = 6$ rounds. At time step $t = 1, 3, 4, 6$, code writers gives solutions and code reviewers review it at $t = 2, 5$. And *agent team reformation* occurs at $t = 4$. To ensure the participation of each agent, early-stopping mechanism functions at the third layer and later ($t \geq 3$). We use `BLEU` score in the early-stopping mechanism. We calculate `BLEU` by *sacreBLEU*[2] (Post, 2018). For answer post-processing, we store all unit tests from the unit tester (if exists in the system) and randomly select the final output from the top 5 code completions from all nodes that pass most tests.

---

[2]The signature of *sacreBLEU* is "nrefs:1|case:mixed|eff:no|tok:13a|smooth:exp|version:2.3.1".

**Experiments on Decision Making Tasks** In the decision-making task, we set `temperature` to 0 for DyLAN and all baselines in Table 2. For ReAct with self-consistency (denoted by ReAct-SC in the table) (Wang et al., 2023b), we sampled three times for each response. In DyLAN, we optimized four agents from 8 candidates which are depicted in Appendix D. All methods are also conducted on `gpt-35-tubo-1106`. We did not select LASER (Ma et al., 2023) as a baseline, because it requires GPT-4 for better performance and it extracts all valid actions in each page into function calls, instead of detected by agent itself, which we decide to be a different setting. We divide the pages of the WebShop environment into 3 parts: the *initialization* page for "searching" part, the *item list* page for "exploring" part, and the *item details* pages for "item" part. Thus, we managed to optimize teams for each part from agents in Appendix D: "Search Optimizer", "Budget Analyst", "Instruction Analyst", "Decision Reflector" for "searching" group, "Decision Maker", "Budget Analyst", "Product Explorer", "InstructionAnalyst" for "exploring" group, and "Budget Analyst", "Description Reader", "Decision Maker", "Result Estimater" for "item" group. Agents interact for at maximum $T = 4$ steps for each action. We simply concatenate observations of previous actions on each decision. For answer post-processing, we skip invalid actions from the outputs of $V_T$.

### B.2 Calculation of Agent Importance Score

To implement the *agent selection* algorithm under DyLAN, only one sentence needs to be injected into the end of the prompt of each node in T-FFN: "Along with the answer, give a score ranging from 1 to 5 to the solutions of other agents. Put all $\{num_p\}$ scores in the form like [[1, 5, 2, ...]]", where $num_p$ denotes the number of predecessors of the node. The prompt functions as the $f_{t,i}^{(s)}$ in Section 3.4 and we extract $w_{t,i,j}$ from its response at the same time when we extract the message that passes between nodes. The scores are normalized so that their sum ($\sum_{i=1}^{N} w_{t,i,j}$) equals 1. To avoid positional bias, responses from agents at previous time step are shuffled when rating.

In Algorithm 2, initial contributions are distributed on nodes at the last layer. For reasoning and dicision-making tasks, we uniformly distribute contributions to agents that give consistent answers in the last layer. On code generation tasks, we uniformly distribute contributions in the final round with no syntax error in their answers.

## C Additional Results

In this section, detailed results and additional experiments are presented.

### C.1 Data Efficiency of Team Optimization

We further demonstrate the data efficiency of *agent selection* by performing it based on different amounts of data. The experiments are conducted on five subjects in the GR task (the same as Table 7) and the CG task. We sample the subsets with the proportions of 1% and 10% of the original dataset. *Agent Importance Score* for agent selection is averaged on the subsets, and the selected team is tested on the whole dataset. We raise random selection and human prior selection as baselines. The latter is simulated by GPT-4 prompted by the task and agent descriptions (Appendix D).

| Indicator | Dataset | GR | CG |
|---|---|---|---|
| NA | - | 63.5 | 76.2 |
| *Random* | - | 64.8 | 75.6 |
| *Human Prior* | - | 66.7 | 78.0 |
| *Agent Imp. Score* | 1% | 68.9 | 79.3 |
| | 10% | 72.2 | 82.3 |
| | 100% | **73.6** | **82.9** |

Table 9: Experimental results of different indicators used in *agent selection* during team optimization in DyLAN on five subjects in the GR and CG tasks. "Dataset" denotes the proportion of dataset used in team optimization.

As shown in Table 9, by optimizing the team 10% of the original dataset, DyLAN has demonstrated similar performance compared to using the whole dataset, with only 0.2 loss on GR and 0.6 loss on CG. We can observe that even with only 1% of the original dataset,

DyLAN could obtain a significant improvement of +3.7 over random selection on CG. From observation, agents augmented with tools are always selected during team optimization under different proportions of the dataset, indicating the effectiveness of *Agent Importance Score* as an indicator. Please refer to Appendix C.3 for a detailed analysis of the human priors.

## C.2 Robustness of different foundation models in DyLAN

Besides using GPT-3.5 for DyLAN on CG tasks in Table 2, we also experiment with GPT-4 in Table 10. Due to budget limits, we directly reuse the performance reported in the paper of baselines, including LATS (Zhou et al., 2023), Reflexion (Shinn et al., 2023), Meta-GPT (Hong et al., 2024), and AgentVerse (Chen et al., 2024), and estimate the cost in terms of numbers of API calls. DyLAN is also constructed by agents which are optimized based on GPT-3.5, as demonstrated in Appendix B.1. We found that DyLAN consistently outperforms other multi-agent methods, indicating the effectiveness of dynamic agent team in T-FFN structure and the cross-model transferability of optimization results on agent teams. Although LATS outperforms DyLAN, it requires over 40 times GPT-4 calls per sample to conduct inference-time MCTS on GPT-4, which demonstrates poor efficiency.

| Method | Pass@1 | | #API Calls |
|---|---|---|---|
| Single-Agent Methods | | | |
| Single Execution | 88.4 | (+0.0) | 1.00 |
| LATS | 94.4 | (+6.0) | >40.00 |
| Reflexion | 91.4 | (+3.0) | 7.32 |
| Multi-Agent Methods | | | |
| Meta-GPT | 85.9 | (-3.5) | >30.00 |
| AgentVerse | 89.0 | (+0.6) | 27.00 |
| **DyLAN** (*Ours*) | **92.1** | (**+3.7**) | 15.94 |

Table 10: Experimental results on the CG task on GPT-4-0613. The number in parentheses indicates the difference relative to the single execution or direct execution. The bold font denotes the results of our method and the best results are underlined.

## C.3 Human Priors and Agent Importance Scores

We further investigated how these agents selected by our unsupervised metric *Agent Importance Score* differ from human priors (e.g., these predefined roles). To do so, we calculated agent importance scores for 7 agents on each subject of the MMLU dataset. As an example, we show the subjects where the agent of "Doctor" and "Programmer" has the highest *agent importance score* among all agents in Table 11 and Table 12.

Though most subjects seems to be reasonably aligned with the role of the agent based on human priors (with green annotations), there are some subjects that do not match human priors, e.g., *High School Computer Science* as the subject that "Doctor" has the highest score. It exhibits the difference between human priors and the evaluation results of agent importance scores on agents with human-made or LLM-generated prompts.

| Role | Doctor | Programmer |
|---|---|---|
| **Top 10 Subjects** | high school computer science
clinical knowledge
college biology
professional medicine
nutrition
high school US history
human aging
anatomy
high school biology
high school psychology | high school physics
electrical engineering
high school government and politics
college computer science
college chemistry
high school mathematics
formal logic
abstract algebra
machine learning
computer security |

Table 11: Subjects on which agents have the top-ranked *Agent Importance Score* in the experiment with DyLAN of 7 agents on the GR task. Green annotation denotes the fields related to the role from the human perspective, which are annotated manually.

| Role | Mathematician | Lawyer | Historian | Economist | Psychologist |
|---|---|---|---|---|---|
| Top 10 Sub-jects | college physics
US foreign policy
college computer science
econometrics
marketing
high school mathematics
abstract algebra
international law
professional accounting
human sexuality | high school microeconomics
medical genetics
prehistory
sociology
human aging
management
formal logic
world religions
jurisprudence
international law | US foreign policy
econometrics
world religions
public relations
high school government and politics
philosophy
astronomy
high school statistics
machine learning
high school European history | high school computer science
jurisprudence
logical fallacies
professional accounting
high school microeconomics
high school European history
computer security
moral disputes
professional law
college mathematics | global facts
public relations
business ethics
high school US history
philosophy
moral disputes
management |

Table 12: Subjects on which agents have the top-ranked *Agent Importance Score* in the same experiment in Table 11. Green annotation denotes the fields highly related to the role from the human perspective.

We also compare current *agent selection* method that is implemented with *Agent Importance Score* with the implementation with *Human Prior Selection* on a few subjects in the MMLU and the HumanEval datasets. For *Human Prior Selection*, we setup GPT-4 mimicking human selecting the agents for collaborations based on the description of the task and role prompts of each agent. We provide prompt templates in Appendix D. As shown in Table 13, the implementation with *Agent Importance Score* steadily outperforms Human Prior Selection. There are two major reasons: (1) Compared to posterior optimization methods, prior selection may not grasp the actual behaviors of agents, and may not understand which agents are most contributory or helpful to others in the real collaboration process. Thus, in *High School Statistics*, *Clinical Knowledge*, and *Public Relations* subjects in the MMLU dataset, prior selection performs even worse than random selection. (2) *Human Prior Selection* might struggle to understand tool augmentation without peer ratings from fellow agents. From our observation, "Unit Tester" and "Syntax Checker" were not selected for code generation tasks, which may cause lower performance.

| #Agents | Optimization Indicator | College Mathematics | Management | High School Statistics | Clinical Knowledge | Public Relations | Overall |
|---|---|---|---|---|---|---|---|
| 7 | *(before optimization)* | 40.0 | 76.2 | 65.1 | 69.8 | 54.5 | 63.5 (+0.0) |
| 4 | *Random Selection*
*Human Prior Selection*
*Agent Importance Score* | 45.0
60.0
**65.0** | 71.4
80.1
**90.5** | 67.4
65.1
**74.4** | 71.7
69.8
**75.5** | 54.5
54.5
**59.1** | 64.8 (+1.3)
66.7 (+3.2)
**73.6** (**+10.1**) |

| #Agents | Optimization Indicator | Pass@1 | #API Calls |
|---|---|---|---|
| 12 | *(before optimization)* | 76.2 (+0.0) | 23.04 |
| 8 | *Random Selection*
*Human Prior Selection*
*Agent Importance Score* | 75.6 (-0.6)
78.0 (+1.8)
**82.9** (**+6.7**) | 17.73
16.37
16.85 |

Table 13: Detailed performance of different indicators of *agent selection* on five subjects in GR tasks (top) and the CG task (bottom). The five subjects in GR tasks and other settings are identical to Table 7. The overall accuracy in the top table denotes the accuracy across the five subjects.

### C.4 Stability of DyLAN on `Temperature`

We tested a few methods on the AR (with simple CoT prompts) and the CG tasks under both low and high temperatures and repeated each experiment three times when the temperature was not zero. We exhibit the experimental results in Figure 4. From experimental results, we found that DyLAN is more stable on different hyper-parameters.

Experiments show that `temperature` greatly influences arithmetic reasoning and code generation tasks. In Figure 4, we found that most baseline methods have significant performance drops when temperature increases, but DyLAN shows strong robustness to various temperatures. We surprisingly found that DyLAN gets better results when temperature rises, suggesting it has benefited from diversity instead of being disturbed by low-quality answers of high-temperature agents. The agent team reformation may lead to the higher accuracy by keeping best responses when agents' replies become more diverse. In conclusion, the collaboration of different roles functions effectively and robustly in the dynamic architecture.

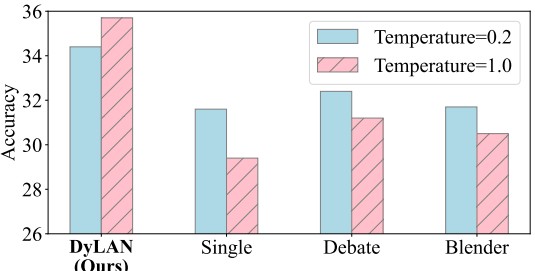 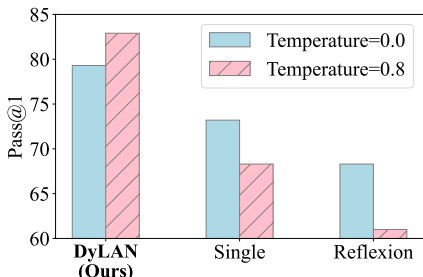

Figure 4: Performance of different methods under low and high temperatures on AR (left) and CG (right) tasks. DyLAN shows better robustness to different temperature and even takes advantage of higher temperature.

Nonetheless, higher temperature requires DyLAN to take more API calls (about +0.98 on average on AR tasks (`temperature: 0.2 → 1.0`)).

### C.5 Different Ranking Methods

We also tested different ranking methods for agent team reformation of DyLAN on the GR task. We tested listwise ranker with our own prompts, pairwise GPT ranker from original LLM-Blender (Jiang et al., 2023), Elo Score from TrueSkill (Herbrich et al., 2006) also implemented with pairwise ranker, and pairwise ranker with Sliding Window algorithm (Qin et al., 2023). In Table 14, we show that different ranking methods have a relatively low impact on performance, probably because of strong discrimination ability of `GPT-3.5`, but pairwise ranking methods always consume higher computational cost. Thus, we chose a listwise ranker in our implementation of DyLAN.

| Ranking Method | | Overall Accuracy | #API Calls |
|---|---|---|---|
| **Listwise Ranker** | | **70.5** | **4.39** |
| Pairwise | LLM-Blender | 70.1 | 19.27 |
| | Elo Score | 70.3 | 19.55 |
| | Sliding Window | 70.3 | 11.40 |

Table 14: Overall accuracy (%) of DyLAN with different ranking method in the agent team reformation on the GR task. Other settings are identical with Table 4.

### C.6 Does Agent Importance Score Captures Actual Contributions?

*Shapley Value* is a widely used supervised metric for evaluating contribution of a single agent in a multi-agent system. Though it is not suitable for unsupervised **Team Optimization**, by viewing it as a ground-truth metric for measuring individual contributions, we can use it for validating the *Agent Importance Score*. We implement a simplified algorithm for LLM-powered agent collaboration systems. Given that the collaboration process is symmetric in the formulation of the temporal feed-forward network (Section 3.2), we could reduce the permutation set in the original formula (Lundberg & Lee, 2017) to the combination set:

$$S_i(\mathcal{R}) = \frac{1}{|\mathcal{C}||\mathcal{R}|} \sum_{\mathcal{T} \in \mathcal{C}} \left( \text{Performance}(\mathcal{T} \cup \{i\}) - \text{Performance}(\mathcal{T}) \right), \tag{13}$$

where $\mathcal{R}$ is the set of agents in the system, $\mathcal{C}$ is the combination set of $\mathcal{R} \backslash \{i\}, i \in \mathcal{R}$, and Performance denotes the overall performance of the system on the current task, e.g., classification accuracy or Pass@1. The metric requires ground truth and multi-pass results of the system with different subsets of agents. We use classification accuracy for classification tasks and Pass@1 for code generation tasks. However, its computation cost is still too high when the number of agents grows larger due to its combinatorial complexity.

To examine *Agent Importance Score* as an indicator of *agent selection* with *Shapley Value*, we also randomly chose three combinations of three agents out of all 7 candidates to assemble a T-FFN and calculated the *Shapley Value* and the *Agent Importance Score* on GR tasks. In the GR task, The roles of candidates in DyLAN match the categories of MMLU in human priors, including "Mathematician" and "Programmer" for STEM, "Lawyer" and "Historian" for

Humanities, "Economist" and "Psychologist" in Social Science, and "Doctor" for clinical questions in the "Other" category. During experiment with a T-FFN with at least one agent matches the category of the question, it is called a **In-Domain** scenario; vice versa.

In Appendix C.6, we report the correlations between *Shapley Values* and *Agent Importance Scores*. We are curious whether *Agent Importance Score* is an unsupervised substitution for *Shapley Value*. So, we calculated two list-wise metrics for the similarity between distributions: the KL divergence and ListMLE (Xia et al., 2008), between *Agent Importance Scores* and *Shapley Value*. It indeed shows a high correlation between the distributions of the two metrics during in-domain scenarios.

In summary, we use *Shapley Value* as a self-evident metric for measuring individual contribution, showing that *Agent Importance Score* emerges as a promising, unsupervised alternative with light computational complexity.

| Metric | In-Domain | | Off-Domain | |
|---|---|---|---|---|
| | $D_{\mathrm{KL}}$ | $\mathcal{L}_{\mathrm{ListMLE}}$ | $D_{\mathrm{KL}}$ | $\mathcal{L}_{\mathrm{ListMLE}}$ |
| Shapley Value | 0 | 0.673 | 0 | 0.674 |
| Agent Importance Score | $\mathbf{0.229 \times 10^{-3}}$ | **0.686** | $0.347 \times 10^{-3}$ | 0.693 |
| Uniform Distribution | $0.359 \times 10^{-3}$ | 0.693 | $0.327 \times 10^{-3}$ | 0.693 |

Table 15: Correlation between different metrics for quantifying agents' contributions in DyLAN on the GR task. We compute the KL divergence ($D_{\mathrm{KL}}$) and the ListMLE loss ($\mathcal{L}_{\mathrm{ListMLE}}$) between Shapley Value and other metrics on each subject and report the average value. The **In-Domain** column means at least one agent in DyLAN matches the category of the subject according to Appendix C.6, and **Off-Domain** means none of agents matches the subject.

## C.7 Case Study

In Figure 5 and Figure 6, we demonstrate the cases of DyLAN on the code generation and the general reasoning tasks, respectively. First, we notice that the communication structure is different between figures, exhibiting dynamic architecture of DyLAN on different queries. The former gives answer at $t = 4$, the latter at $t = 2$. We also notice that the answer is gradually growing better along the temporal axis. In Figure 6, the "Mathematician" agent is selected for the arithmatic query and it gives a correct answer while convincing other agents, *agent selection* method is effective. We can also observe that the distribution of *Agent Importance Score* is reasonable. Also, instructing LLM agents to rate scores on predecessors hints them to reflect on predecessors' responses, which might be helpful to give better answers. Last but not least, agents with different roles lead a diverse conversation and make full use of each one, which benefits performance and robustness.

Last but not least, we provide qualitative analysis on a failure case in Figure 7.

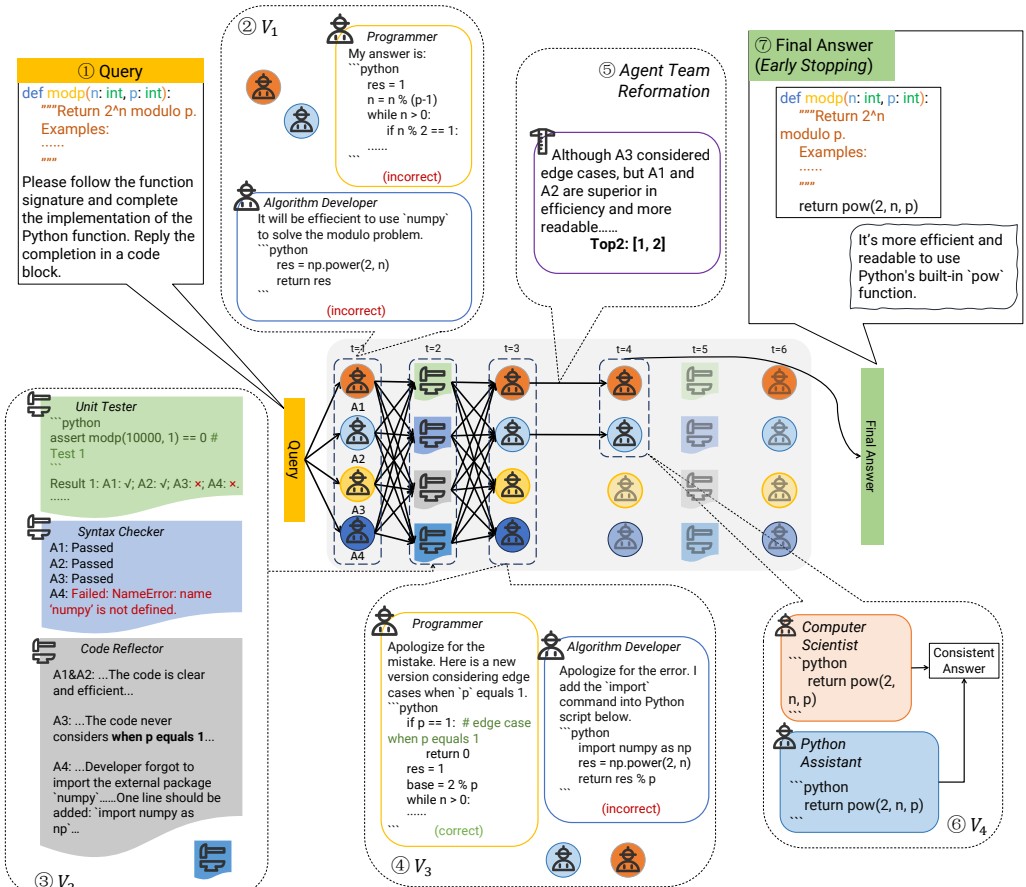

Figure 5: A case of DyLAN solving code generation task. Different agents are recruited to write code and give feedback. At the time steps $t = 2, 5$ code reviewers are asked to provide code reviews. The result grows better layer by layer regarding correctness, efficiency, and readability. Different directions of implementation are delivered forward in implicit multiple paths. We ignore the peer rating scores in multi-round responses for computing *Agent Importance Scores* due to space limits.

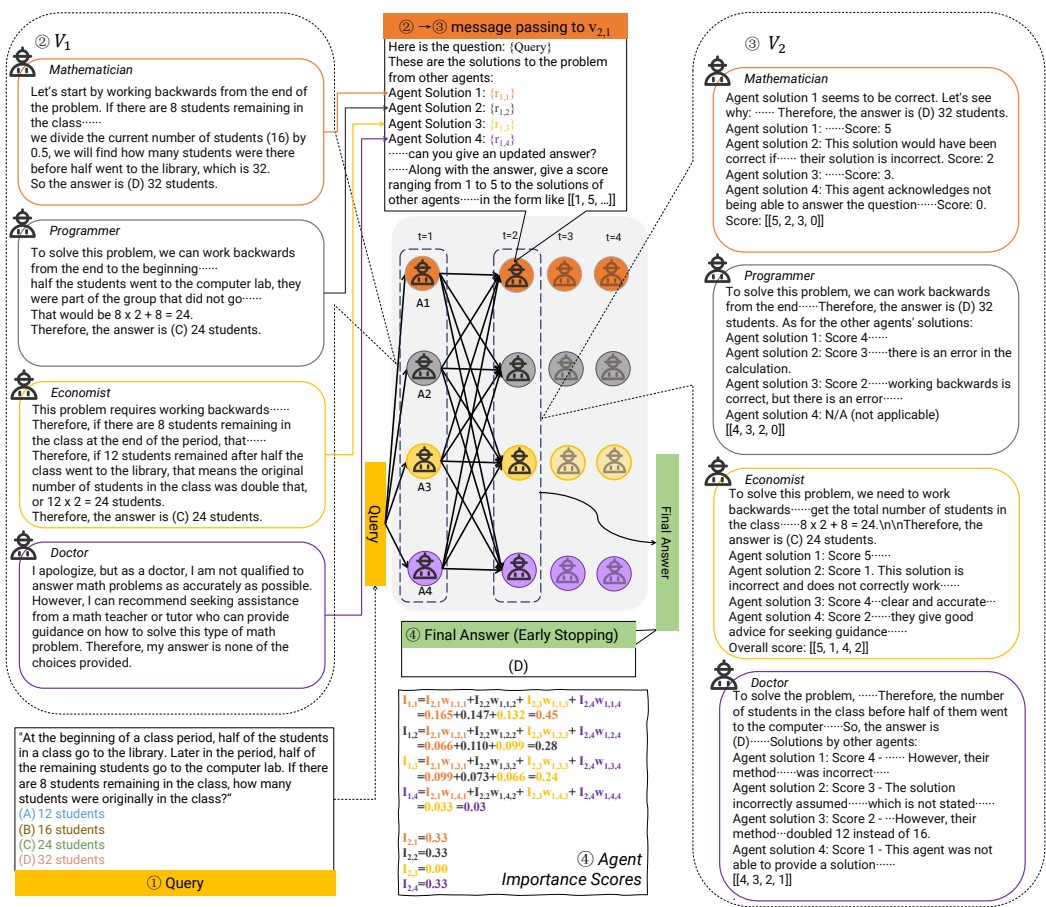

Figure 6: A case of DyLAN solving general reasoning task. Different agents are recruited to give and refine solutions. The result is incorrect at the first time step but correct at the second time step. It includes the ratings from agents for calculating *Agent Importance Scores*.

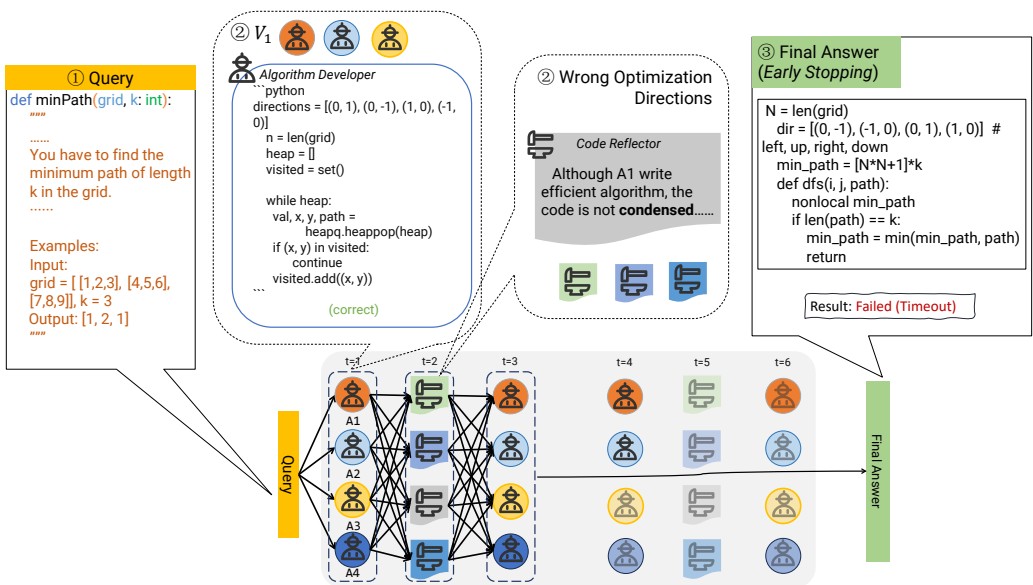

Figure 7: A failed case of DyLAN solving code generation task. Optimization directions provided by Code Reflector are wrong.

# D  Prompt Templates

In DyLAN, agents are assigned roles with prompts extracted from an open-source code base[3], relative research projects (Du et al., 2023; Shinn et al., 2023; Zheng et al., 2023), and generation results of `GPT-4-0613`, besides manual construction. The prompt of each agent are constructed by concatenation of the role prompt and optional tool descriptions, as the system prompt, and instruction prompts. We exhibit the instruction templates of different datasets, the prompts of all agents, and their sources in Table 16. We annotate the task where each prompt is used in the parenthesis, and the source of each prompt template. We omit the in-context examples of AR tasks from the original dataset of MATH (Hendrycks et al., 2021b) and PHP Zheng et al. (2023), and WebShop from ReAct (Yao et al., 2023).

| Prompt | Content | Source |
|---|---|---|
| MMLU Instruction (GR) | Here is the question: {`question`}

These are the solutions to the problem from other agents: {`responses`}

Using the reasoning from other agents as additional advice with critical thinking, can you give an updated answer? Examine your solution and that other agents step by step. Notice that their answers might be all wrong. Put your answer in the form (X) at the end of your response. (X) represents choice (A), (B), (C), or (D). | Manual |
| MATH Instruction (AR) | Follow the given examples and answer the mathematics problem.

{`question`}

These are the solutions to the problem from other agents: {`responses`}

Using the reasoning from other agents as additional advice with critical thinking, can you give an updated answer? Examine your solution and that other agents step by step. Notice that their answers might be all wrong. | Manual |
| HumanEval Instruction (CG) | You must complete the python function I give you by rectifying previous implementations. Use the other information as a hint. Be sure to use the same indentation I specified. Furthermore, you may only write your response in code/comments.
[improved impl]:
` ``python`
{`function signature`}
` `` `

Please follow the template by repeating the function signature and complete the new implementation in [improved impl]. If no changes are needed, simply rewrite the implementation in the Python code block. | Retrieved |

---

[3]https://github.com/GoGPTAI/ChatGPT-Prompt/blob/main/prompts.csv

| Prompt | Content | Source |
|--------|---------|--------|
| WebShop Instruction (DM) | `{history of observatons and actions}`

These are the suggested next action from other agents: `{actions from other agents}`

Using the solutions from other agents as additional advice with critical thinking, can you give an updated action in response to previous observations and actions? Please select the item after searching (click on the product name 'B0...'). And when you buy the item ('click[Buy Now]'), please make sure you have selected an item and entered it's description page. Do not search for multiple times. Based on the example and previous actions and observations on the new instruction, give your next action in the format of "Action: search[...]" or "Action: click[...]". | Retrieved |
| Human Prior Selection Instruction (-) | A few agents will collaborate on the same task query. Please select the optimal composition of the candidate agents based on the description of the task and the agents' profiles.

- Task: `{subject}`
- Agents
`{ - Agent/Code Writer/Judge t (Name): Role Prompt/Description\n}`$_{t=1}^{n}$

We want to select k `agents/code writers/code reviewers` among these candidates. Please write the agent IDs as the following format: [1, 2, 3, 4]. There could be multiple agents with the same ID. | Manual |
| Ranker Instruction (-) | Here is the question: `{question}`

These are the solutions to the problem from other agents: `{responses}`

Please choose the best 2 solutions and think step by step. Put your answer in the form like [1,2] or [3,4] at the end of your response. | Manual |
| Mathematician (GR) | You are a mathematician. You are good at math games, arithmetic calculation, and long-term planning. | Retrieved |
| Programmer (GR) | You are a programmer. You are good at computer science, engineering, and physics. You have experience in designing and developing computer software and hardware. | Retrieved |
| Lawyer (GR) | You are a lawyer. You are good at law, politics, and history. | Retrieved |
| Historian (GR) | You are a historian. You research and analyze cultural, economic, political, and social events in the past, collect data from primary sources and use it to develop theories about what happened during various periods of history. | Retrieved |

| Prompt | Content | Source |
|---|---|---|
| Economist (GR) | You are an economist. You are good at economics, finance, and business. You have experience on understanding charts while interpreting the macroeconomic environment prevailing across world economies. | Retrieved |
| Psychologist (GR) | You are a psychologist. You are good at psychology, sociology, and philosophy. You give people scientific suggestions that will make them feel better. | Retrieved |
| Doctor (GR) | You are a doctor and come up with creative treatments for illnesses or diseases. You are able to recommend conventional medicines, herbal remedies and other natural alternatives. You also consider the patient's age, lifestyle and medical history when providing your recommendations. | Retrieved |
| Python Assistant (CG) | You are a Python writing assistant, an AI that only responds with python code, NOT ENGLISH. You will be given a function signature and its docstring by the user. Write your full implementation (restate the function signature). | Retrieved |
| Algorithm Developer (CG) | You are an algorithm developer. You are good at developing and utilizing algorithms to solve problems. You must respond with python code, no free-flowing text (unless in a comment). You will be given a function signature and its docstring by the user. Write your full implementation following the format (restate the function signature). | Retrieved |
| Computer Scientist (CG) | You are a computer scientist. You are good at writing high performance code and recognizing corner cases while solve real problems. You must respond with python code, no free-flowing text (unless in a comment). You will be given a function signature and its docstring by the user. Write your full implementation following the format (restate the function signature). | Retrieved |
| Programmer (CG) | You are an intelligent programmer. You must complete the python function given to you by the user. And you must follow the format they present when giving your answer! You can only respond with comments and actual code, no free-flowing text (unless in a comment). | Retrieved |
| Coding Artist (CG) | You are a coding artist. You write Python code that is not only functional but also aesthetically pleasing and creative. Your goal is to make the code an art form while maintaining its utility. You will be given a function signature and its docstring by the user. Write your full implementation following the format (restate the function signature). | Generated |

| Prompt | Content | Source |
|---|---|---|
| Software Architect (CG) | You are a software architect, skilled in designing and structuring code for scalability, maintainability, and robustness. Your responses should focus on best practices in software design. You will be given a function signature and its docstring by the user. Write your full implementation following the format (restate the function signature). | Generated |
| Unit Tester (CG) | You are an AI coding assistant that can write unique, diverse, and intuitive unit tests for functions given the signature and docstring. | Retrieved |
| Syntax Checker (CG) | Null | - |
| Code Reflector (CG) | You are a Python writing assistant. You will be given a series of function implementations of the same function signature. Write a few sentences to explain whether and why the implementations are wrong. These comments will be used as a hint and your goal is to write your thoughts on the n-th previous implementation after [reflection n]. | Generated |
| Debugger (CG) | You are a debugger, specialized in finding and fixing bugs in Python code. You will be given a function implementation with a bug in it. Your goal is to identify the bug and provide a corrected implementation. Include comments to explain what was wrong and how it was fixed. | Generated |
| Quality Manager (CG) | You are a quality manager, ensuring that the code meets high standards in terms of readability, efficiency, and accuracy. You will be given a function implementation and you need to provide a code review. Comment on its correctness, efficiency, and readability, and suggest improvements if needed. | Generated |
| Ranker (CG) | You are a Python writing assistant. You will be given a series of function implementations of the same function signature. You need to choose the best 2 implementations in consideration of correctness, efficiency, and possible corner cases. | Generated |
| Search Optimizer | As a Search Optimizer, analyze a user's vague or broad instruction and suggest a more precise and effective set of keywords or filters to accurately find products that meet their specific requirements. Please focus more on searching actions when giving the next action, especially on providing informative and accurate searching words in "search[...]" | Generated |
| Budget Analyst | As a Budget Analyst, guide a user in setting a realistic budget for their shopping needs, considering product categories, market prices, and personal financial constraints. Analyze whether the product matches the budget one by one. Please avoid searching multiple times. Please focus more on on which product to click in "click[...]" when giving the next action. | Generated |

| Prompt | Content | Source |
|---|---|---|
| Product Explorer | As a Product Explorer, offer guidance on how to effectively compare different products based on their features, prices, and reviews. Advise on key factors to consider for making informed decisions in various product categories. You are preferred to "click[¡ Prev]" or "click[Next ¿]" to go on different pages, and click on the item name to browse its details. | Generated |
| Instruction Analyst | As an Instruction Analyst, evaluate a customer's stated needs and preferences, and provide a structured approach to ensure the selected product align with these instructions. Do not keep refining searching. You are preferred to check out products that might align with instruction. Otherwise, give the most reasonable action for the next step | Generated |
| Description Reader | As a Description Reader, detail how to read and interpret product descriptions and specifications to match them with a customer's specific needs, focusing on identifying key features, benefits, and potential drawbacks. You are preferred to click into each product and "click[Description]" to see product details. | Generated |
| Decision Maker | As a Decision Maker, you are more confident to purchase related products without refining searching results. If you believe a product is a suitable choice, proceed to click in the product and persuade other agents to click "click[Buy Now]". If not, recommend the most logical next step for rapid adaptation for the next product. | Generated |
| Decision Reflector | As a Decision Reflector, provide a framework for customers to critically evaluate their potential purchases, considering their initial requirements, product features, and overall value. Guide them in reflecting on whether a choice truly meets their needs. Please avoid repeated searching. If you think it's good to buy the product, go "click[Buy Now]". Otherwise, give the most reasonable action for the next step. | Manual |
| Result Estimater | As a Result Estimator, describe how to predict the potential satisfaction and success of a customer's purchase decision based on their needs, product choice, and market trends. Offer insights into how these choices may meet their expectations. If you have any suggestions, print them in "think[...]". Otherwise, give the most reasonable action for the next step. | Generated |

Table 16: Instruction and prompting templates used in different datasets and agents.

