# OpenReview forum: "A Dynamic LLM-Powered Agent Network for Task-Oriented Agent Collaboration"
_colmweb.org/COLM/2024/Conference — COLM_

### Official Review · Reviewer_sGnL · 2024-05-10

**Rating:** 6
**Confidence:** 3
**Ethics Flag:** 1

**Summary:**

This paper introduces an approach called DyLAN that incorporates a two-stage paradigm: i) Team Optimization, the temporal feed-forward neural network (T-FFN) that selects most contributary agents for the given task and ii) Task solving, the T-FFN facilitate agents to communicate with each others to solve a bunch of a diverse of tasks. To further enhance the efficiency, DyLAN uses early stopping mechanism to terminate the conversation when most of agents in a single layer agree to have a consistent answer.  In the experiment section, the DyLAN is proved to outperform with baselines such as LLM-Blender, LLM Devate, ReAct, and Reflexation, etc, on code generation, decision making, and general reasoning, etc, and shows that DyLAN.

**Reasons To Accept:**

1. Solid Experiments: The paper provides a robust experimental methodology, demonstrating DyLAN's effectiveness across a range of downstream tasks such as coding, general knowledge quetsion answering, math etc. It shows that cost DyLan can enhance the performance of the above mentioned tasks while keep low cost on computation with respect to reasonable times of function call.

2. The idea of DyLAN is novel, it trains a dynamic temporal feedforward network to select the imporatant agents, and the early stopping seems to be effective to reduce the cost on long-term conversation.

3. The ablation study is thorouh.

**Reasons To Reject:**

1.  **Extended Query on the Selection and Performance of Agents in the Initial Pool**: "Upon close examination of Figure 5 within the manuscript, there appears to be a lack of clarity regarding the roles adopted by the other agents involved in the code generation task. It is noted that the 'programmer' and 'algorithm developer' agents seem to outperform the rest, which might suggest that these two agent roles are disproportionately influential in the task's outcome. This observation raises the question of whether these two agent roles alone suffice for the task at hand. Similarly, in the mathematical reasoning tasks, the 'programmer' and 'mathematician' agents are indicated as the primary contributors, prompting the inquiry of whether including agents with roles such as 'doctor' or 'economist' might be an overcomplication in the context of mathematical tasks. If the initial agent pool is curated for specificity to different tasks, then the inclusion of seemingly irrelevant agents could be considered redundant and may detract from the efficiency of the overall system. Could the authors provide insight into the rationale behind the selection of these varied roles and how their inclusion impacts the performance and generalizability of the DyLAN framework?"

2. Upon reviewing Figures 5 and 6, it is observed that layers with an even number typically assume a critical function, evaluating the responses generated by the preceding odd-numbered layers. This pattern suggests that the effectiveness of the DyLAN framework may be contingent upon the utilization of highly curated prompts tailored to fit each agent's distinct roles and scenarios. The observed phenomenon raises concerns regarding the scalability and flexibility of the system, particularly in scenarios where the diversity of tasks and roles is substantial. It would be beneficial for the authors to address the necessity for such detailed prompt curation and discuss potential strategies to streamline this process, ensuring that the framework remains adaptable across a wide array of tasks without compromising performance.

However, I find the paper to be of high quality overall; if the authors can provide clear and satisfactory responses to the queries raised, I would be willing to increase the points.

---

> ### Author Rebuttal · Authors · 2024-05-27
>
> We sincerely thank the reviewer for the feedback.
> # 1. Selection of Agents
> - **Selecting a Proper Team is Challenging:** We acknowledge that some agents might be redundant for specific queries. Empirically, fewer properly selected agents can outperform more with redundancy (Table 5&Fig.3). **However, identifying these agents precisely is hard.** Our goal is to identify the most contributory agents in a principled way. The Agent Importance Score is effective as an unsupervised metric, but not an oracle.
>   - **The Challenge to Identify Contributory Agents:** In Fig.5, agents do not reach the correct answer at t=1 but later. This illustrates the difficulty of agent identification brought by multi-turn communications. Also, **seemingly irrelevant agents from human perspectives may make significant contributions** (App.C.5). Empirically, Table 13 shows selecting agents based on human preference performs lower.
>   - **The Challenge to Decide the Team Size:** In Fig.6, agents refine their answers after talking to others. This case illustrates the challenge to determine the optimal team size priorly. Although Mathematician alone is sufficient for the query in Fig.6, it is not always adequate for others. In Fig.3 and Table 8, the chosen team size in our settings strikes better effectiveness and efficiency.
> - **Generalization Benefits:** Including various roles could enhance generalization. In Table 10, different subjects benefit from distinct selected teams. But if the majority are irrelevant to tasks by design (e.g., coding agents for DM tasks), the result may be negatively impacted.
> # 2. Framework Design and Scalability
> - **General Formulation:** The pattern of DyLAN is designed to be general. DyLAN treats each agent at each time step equally (Sec.3.2). In Fig.6, the agents at t=2/4 are identical to those at t=13. The pattern is consistent in AR and DM tasks.
> - **Role of Even-Numbered Layers:** In Fig.5, judges are in the even-numbered time steps because they require predecessors. The setup is of the general pattern.
> - **Generalizability and Scalability Across Tasks:** For instance, in GR tasks with 57 different subjects, DyLAN selects teams from 7 varied agents, significantly improving performance across subjects (Table 5&10). The system prompt for each agent is universal and prompt in DyLAN for various tasks is only different in task instructions (usually 1 sent.).
>
> ---
> **We appreciate your time in reading our rebuttal and look forward to further discussions.**

---

> > ### Comment · Reviewer_sGnL · 2024-06-05
> > **Response**
> >
> > Thank you for the response. I keep the score (6) unchanged.

---

### Official Review · Reviewer_B67Q · 2024-05-11

**Rating:** 8
**Confidence:** 3
**Ethics Flag:** 1

**Summary:**

In this paper, the author introduce the Dynamic LLM-Agent Network (DyLAN) framework, designed to facilitate collaboration automatically among multiple LLM agents.The author employs T-FNN to dynamically regulate interactions between different agents and Introducing Agent Importance Score to measure each agent's contribution, reducing manual intervention, which is quite innovative in this field. The whole framework includes two stages: "Team Optimization", which identifies and excludes underperforming agents, and "Task solving" performing inference based on the T-FFNs.

The author conducted experiments on DyLAN across four tasks: Code Generation, Decision Making, General Reasoning, and Arithmetic Reasoning, comparing it with several previous SOTA methods. DyLAN not only improves performance but also keeps API calls at a relatively low level. To demonstrate the effectiveness of DyLAN, the author conducted lots of experiments and discussions. However, many important findings were included in the appendix. Personally, I believe some crucial discussions such as C.2 and C.8 should be included in the main paper.

**Questions To Authors:**

1. In the ablation study, I noticed an interesting phenomenon: the early stop has a significant impact on API calls in arithmetic reasoning and general reasoning tasks. Whereas in the decision-making task, both methods have considerable impacts. For code generation, however, neither method seems to have much effect. What reasons might the author attribute to this phenomenon?

**Reasons To Accept:**

1. Using T-FNN to dynamically regulate interactions between different agents and Introducing Agent Importance Score to measure each agent's contribution, reducing manual intervention, which is quite innovative in this field.

2. The idea of introducing Byzantine Consensus theory to achieve early stopping is very interesting and proved to be effictive through ablation study.

3. The author demonstrated the outstanding performance of DyLAN through various experiments and comparisons with current sota methods.

**Reasons To Reject:**

I don't have any major concerns that would lead me to reject the submission of this article, although:
1. As I said  in  summary, some crucial discussions should be included in main paper.
2. The improvement of DyLAN on the AR task is limited compared to other tasks. What might cause this phenomenon?

---

> ### Author Rebuttal · Authors · 2024-05-27
>
> We sincerely thank the reviewer for the constructive feedback. Below, we address each of the concerns you raised.
>
> # 1. Crucial Discussions in Main Paper
>
> We appreciate your observation and agree that some crucial discussions should be included in the main paper. We will consider incorporating these discussions for a more comprehensive view of our work within the page limits.
>
> # 2. Limited Improvement on the AR Task
>
> We have identified the following reasons for the phenomenon:
>
> - **Knowledge Dependency of MATH Dataset:** The AR task utilizes the challenging MATH dataset, which is highly knowledge-dependent. Despite this, DyLAN still significantly outperforms baselines under the same prompting strategy, with relative improvements of 10.2% and 3.0% compared to the best baseline, as shown in Table 2.
> - **Role Assignment:** Compared to other tasks, role-playing introduces relatively lower improvements in AR tasks, due to the high knowledge dependency of MATH. In comparison, Table 13 shows that selecting appropriate agents for math subjects in MMLU, which are less knowledge-dependent, results in greater performance improvement.
>
> # 3. Impact of Early Stopping and Agent Team Reformation
>
> We explain the reason for phenomena for each task:
>
> - **GR and AR Tasks:** Early stopping functions when answers from agents are exactly the same. For multiple-choice problems in GR or mathematical problems, we use exact matching to compare answers. Early stopping is more effective in GR since problems are usually simpler than AR, allowing agents to reach consensus earlier.
> - **CG Tasks:** Answer comparison is more challenging for open-ended tasks like CG tasks. We use the BLEU score with a 0.9 threshold for consistency checks. This makes early stopping less effective since agents may generate codes in different formats, leading to fewer opportunities to stop early.
> - **DM Tasks:** DM tasks face a similar challenge due to the large action space and arguments (e.g., click[item name], search[query words]). The multi-step nature of DM tasks means that the length of actions for each sample is greatly affected by the quality of decisions. With incorrect actions, the agent may continue acting until the maximum step is reached, making both ablations significantly impactful.
>
> We will include these discussions in the next version of our manuscript to provide a clearer understanding.
>
> ---
>
> **We appreciate your time in reading our rebuttal and look forward to further discussions.**

---

### Official Review · Reviewer_2JJf · 2024-05-20

**Rating:** 6
**Confidence:** 3
**Ethics Flag:** 1

**Summary:**

In this paper, it proposes a new method for dynamically conduct multi-agent collaboration. In incorporates the team operation process and task solving process, which can improve the performance of agent collaborations. The experiments have shown the effectiveness of this method.

**Questions To Authors:**

see above

**Reasons To Accept:**

Advantage:
1.The problem is practical and the motivation is sensible. It is similar to create a meta agent filter to construct an agent team. Some previous works like MetaGPT also study this problems as well.
The techniques sound great, which utilizes methods from graph learni

**Reasons To Reject:**

Shortcoming & Question:
1.It seems that there is no error bar in the results of experiments.
2.What about the comparison with MetaGPT?
3.The paper formatting can be improved, such as reducing the empty blocks within it.

---

> ### Author Rebuttal · Authors · 2024-05-27
>
> We sincerely thank the reviewer for the constructive feedback. Below, we address each of the concerns you raised.
> # 1. Error Bars in Experimental Results
> Thank you for pointing out the absence of error bars in our results. We acknowledge the importance of including error bars for a comprehensive presentation. There are two main reasons for this omission:
>
> 1. **Variance Related to “Temperature”:** The variance in our results is highly influenced by the hyper-parameter “temperature,” which is searched within a range (e.g., {0, 0.2, 0.8, 1.0} for GR&AR), as detailed in Appendix B.1. Different temperatures were used for different methods. Therefore, the variance mainly reflects the impact of temperature rather than providing additional insightful information about the method.
>
> 2. **Monetary Constraints:** Due to budget limitations, we conducted three runs for methods with non-zero temperatures. Three samples were insufficient to calculate meaningful error bars (e.g., 1-sigma or 2-sigma bars). Consequently, we opted to report the median result, which offers a reliable representation within our resource constraints.
> # 2. Comparison with MetaGPT
> **Performance:** Please refer to Table 7 in Appendix C.3. Our findings indicate that MetaGPT (85.9) performs worse than bare GPT-4-0613 (88.4) with task instructions and requires significantly more API calls for each sample in code generation tasks. DyLAN (92.1) achieves 7.2% improvement against MetaGPT. Additionally, MetaGPT's framework is not easily adaptable to non-programming tasks.
>
> **Methodology:**
> - **Agent Filtering:** MetaGPT does not incorporate agent filtering methods. Our approach to team optimization involves selecting a team of agents from an initial pool, which is distinct from MetaGPT's fixed framework with five predefined agent roles.
> - **Related Work:** As further related work, SPP [1] introduces agent generation by LLMs but does not filter agents. The exchange-of-thought [2] employs thought-level consistency for response filtering, which is not applicable at the agent level. Currently, we are not aware of any principled agent-level team optimization methods besides DyLAN.
> # 3. Paper Formatting
> Thank you for your feedback regarding the paper's formatting. We will make necessary formatting adjustments to improve our paper.
>
> [1] https://arxiv.org/abs/2307.05300.
> [2] https://arxiv.org/abs/2312.01823.
>
> ---
>
> **We appreciate your time in reading our rebuttal and look forward to further discussions.**

---

### Decision · Program_Chairs · 2024-07-10

**Decision:**

Accept

**Comment:**

This paper introduces a new framework, DyLAN, for dynamic selection and collaboration of LLM-powered agents tailored to specific tasks and domains. Key strengths include the introduction of an Agent Importance Score for team optimization, which significantly enhances performance over static methods, and the comprehensive empirical validation across various tasks such as code generation, decision-making, general reasoning, and arithmetic reasoning.
The reviewers appreciated the practical motivation and technical soundness of the approach. By examining the reviewers' comments and the author responses, the AC thinks this paper warrants an acceptance with minor improvements in the camera ready version.